# Growth Hormone Replacement Therapy Seems to Be Safe in Children with Low-Grade Midline Glioma: A Series of 124 Cases with Review of the Literature

**DOI:** 10.3390/cancers15010055

**Published:** 2022-12-22

**Authors:** Coline Puvilland, Carine Villanueva, Anaëlle Hemmendinger, Laure Kornreich, Iva Gueorguieva, Mélodie-Anne Karnoub, Pierre Aurélien Beuriat, Pierre Leblond

**Affiliations:** 1Pediatric Endocrinology Unit, Woman Mother Child Hospital, Lyon Civil Hospices, 69500 Bron, France; 2Pediatric Unit, Hospital Fleyriat, 01012 Bourg-en-Bresse, France; 3University Hospital Center of Hôpital Nord, 13015 Marseille, France; 4Department of Hematology and Oncology, Queen Fabiola Children’s University Hospital, 1020 Brussels, Belgium; 5Pediatric Endocrine Unit, Jeanne-de-Flandre Children’s Hospital, 59000 Lille, France; 6Department of Pediatric Neurosurgery, Lille University Hospital, 59000 Lille, France; 7Department of Pediatric Neurosurgery, Woman Mother Child Hospital, Lyon Civil Hospices, 69500 Bron, France; 8Rockfeller School of Medicine, Claude Bernard University Lyon 1, 69100 Villeurbanne, France; 9Department of Pediatric Oncology, Institute of Pediatric Hematology and Oncology (IHOPe), Centre Léon Bérard, 69008 Lyon, France

**Keywords:** growth hormone (GH), growth hormone replacement therapy (GHRT), growth hormone safety, low-grade glioma, tumor relapse, endocrine monitoring

## Abstract

**Simple Summary:**

Growth hormone (GH) deficiency is the most common hypothalamic-pituitary disorder due to a brain tumor during childhood, whether it is related to the tumor itself or to the treatment. Nevertheless, the use of growth hormone replacement therapy (GHRT) remains controversial due to its potential proliferative properties. Few data are available on the safety of growth hormone replacement therapy in children with low-grade gliomas (LGG). The aim of our retrospective study was to assess the impact of growth hormone replacement therapy on the risk of relapse in children treated for midline low-grade gliomas. We included 124 children treated for low-grade midline glioma. Among them, 17 were supplemented with growth hormone. There was no significant difference in terms of relapse between the group supplemented with growth hormone and the group not supplemented. These results support the safety of growth hormone in this population.

**Abstract:**

There is little scientific evidence regarding the safety of GHRT in LGG, where GH deficiency is common. Purpose: to compare the recurrence rate in children with midline LGG, depending on whether or not they have received GHRT, in order to assess its impact on the risk of tumor recurrence. Methods: This bicentric retrospective study included 124 patients under the age of 18 who were diagnosed with a midline low-grade glial tumor between 1998 and 2016. We also reviewed literature on this subject. The main outcome measure was tumor relapse, demonstrated by brain MRI. Results: There were 17 patients in the GH-supplemented group (14%) and 107 patients in the non-supplemented group (86%). Relapse occurred in 65 patients (45.5%); 7 patients died (4.9%); no deaths occurred in patients receiving GHRT. Two patients developed a second tumor (1.4%), none of which had received GHRT. Relapse concerned 36.4% of patients without GHRT and 52.9% of patients with GHRT. The difference was not statistically significant between the two groups (*p* = 0.3). Conclusion: GHRT does not lead to a statistically significant increase in risk of relapse for pediatric midline low-grade pediatric glioma in our cohort. Although these results appear reassuring, future natural history or prospective studies should be done to ascertain these findings. Nevertheless, these reassuring data regarding GHRT are in agreement with the data in the current literature.

## 1. Introduction

Long-term overall survival for children with low-grade glioma (LGG) is now 95% at 5 years [1] and 87% at 10 years [2] thanks to recent scientific advances. The major challenge is to prevent long-term sequelae in what is often a chronic, but rarely fatal, disease. Growth hormone (GH) deficiency is the most common hypothalamic-pituitary disorder due to a brain tumor during childhood, whether it is related to the tumor itself or to the treatment received [3]. Indeed, in the absence of GH replacement therapy (GHRT), 40% of pediatric brain tumor long-term survivors have a short adult height below the tenth percentile [4]. Additionally, GHRT is important to prevent short stature in adulthood, osteoporosis, or metabolic syndrome [5].

GH has a proliferative role, and it inhibits apoptosis by activating the PI3K-AKT-mTOR signaling pathway [6]. This same signaling pathway is known to be involved in the carcinogenesis of certain LGGs in children [6].

Therefore, it is important to assess the risk of tumor recurrence or progression in patients with LGG when GHRT is used, especially in the case of residual tumor. Precautions before starting GHRT are well known: the glioma should be considered inactive and all anti-tumor treatment should have been completed for at least 12 months, as recommended [5,7]. GH treatment should be discontinued if there are signs of tumor growth [8].

The safety of GHRT has already been proven by numerous studies when used in children or adults without specific risk factors for developing cancer [6,8,9]. Several studies have already shown that the risk of tumor recurrence is not increased with GHRT in patients with craniopharyngioma, medulloblastoma, or malignant germ-cell tumor [6,10,11,12,13,14,15,16,17]. Some studies have also investigated gliomas and GHRT and did not show an increased risk of relapse occurring in connection with GHRT [6,12,16]. However, these are conclusions drawn from subgroups of the population. The results are quite disparate from one study to another.

Midline location is a statistically significant independent risk factor for growth hormone deficiency due to proximity to the pituitary and hypothalamus [2,5]. Growth hormone (GH) deficiency is common in this location due to tumor growth, tumor removal, or radiation therapy in the hypothalamic-pituitary region [7]. No studies to date have focused on midline LGGs.

While overall survival is very good, progression-free survival is less than 40% at 5 years [18]. LGG therefore often becomes a chronic disease. Over the long term, overall survival is 50.4% at 18 years for gliomas of the optic pathways in particular [18]. The main cause of death is tumor progression. The neuro-cognitive sequelae are often significant in the long term [19]. Age and intracranial hypertension at diagnosis are often associated with a worse prognosis [18]. Currently, the management of LGG is primarily surgical excision, which can be curative when total excision is possible. Unlike other LGGs in which surgery is often the only treatment (e.g., posterior fossa), surgical removal of midline LGGs is often not possible due to proximity to the optic chiasm brainstem and hypothalamic pituitary axis [20]. Indeed, the risk of visual, neurological, and endocrine sequelae is major [19]. When surgery is impossible or when residual tumor persists after surgery, the risk of progression or relapse is high. In that case, chemotherapy is usually given. The three most commonly used protocols in France during this period were the BBSFOP protocol (6 cytotoxic agents given sequentially for 16 months and including carboplatin, procarbazine, etoposide, cisplatin, vincristine, and cyclophosphamide) [21], the SIOP-LGG 2004 protocol (vincristine carboplatin) [22] and the Vinblastine protocol [23]. Sometimes, radiotherapy remains the most suitable treatment, but the tendency is to use it less, due to radiation-induced later effects, especially in very young children (secondary cancer, post-radiation angiopathy, hypothalamic-pituitary dysfunction, and/or slowed cognitive development) [18].

The main objective of our study was to compare the recurrence rate in children with midline LGG, depending on whether or not they received growth hormone replacement therapy, in order to assess its impact on the risk of tumor recurrence. The secondary objective was to carry out a descriptive, retrospective analysis of the cohort of children treated for a low-grade glial tumor of the midline, between 1 January 1998 and 31 December 2016 at the university hospitals of Lille and Lyon in France.

## 2. Materials and Methods

The study was an observational, retrospective and descriptive study. The information was collected by a standardized and anonymized data collection sheet. In accordance with French regulations, the study protocol was approved by the Commission Nationale de l’Informatique et des Libertés (CNIL) on the basis of a declaration of conformity MR-004 (No. 2215746).

We defined the midline tumors as limited to the diencephalon, optic nerves, optic chiasm, pituitary stalk area, hypothalamus, epiphysis, thalamus, and third ventricle. We included all patients treated for low-grade midline glioma diagnosed between 1 January 1998 and 31 December 2013 in Lille (University Hospital and Center Oscar Lambret comprehensive cancer center) and between 1 January 2000 and 31 December 2016 in Lyon (IHOPe), allowing a minimum of 3 years follow-up at the time of data collection for all patients. Patients had to be between 0 and 18 years of age at the time of diagnosis. We excluded patients for whom follow-up was not carried out only in these centers, patients lost to follow-up, and those for whom there was a lack of data. Patients who died before the end of first-line oncological treatment were also excluded as well as those who had recurrence within one year of the last oncological treatment. One patient was excluded from the series because his GH replacement therapy was started before one year of remission, contrary to the recommendations [10]. We performed a comparative analysis of the tumor recurrence rate between patients substituted and unsubstituted for growth hormone.

The risk factor studied was exposure to growth hormone. For the diagnosis of GH deficiency, IGF1 was measured as well as IGFBP3 in young children. GH must be measured after stimulation in France by hypoglycemia (insulin tolerance test), by L-dopa, arginine, glucagon, propranolol, clonidine, or GHRH [24]. These stimulation tests can be used in combination. The diagnosis of GH deficiency (GHD) must be proven by two tests, a simple stimulation test and a combined stimulation test. If both tests show a result below 10 mIU/mL (3 micrograms/L), it is a severe GH deficiency. There is a partial GH deficiency if the results are between 10 and 20 mIU/L (3 to 6.7 micrograms/L). A single test with a response >6.7 micrograms/L excludes the diagnosis of GHD [24]. If there is an obvious acquired cause of GHD, a single stimulation test measuring IGF1 is necessary [24]. In our study, all patients were monitored for height and weight growth. Growth hormone deficiency was suspected in children presenting with height curve brake and confirmed on IGF1 levels and by two growth hormone stimulation tests.

The primary endpoint was relapse. Relapse was defined by the reappearance of the tumor on imaging (in a patient previously in complete remission), or the progression in volume of a pre-existing remnant tumor, occurring outside of any oncological treatment, or at least one year following the diagnosis for patients who have not received any treatment apart from a possible biopsy. The tumor was considered as stable if it did not increase in volume on two consecutive MRI scans, one year apart, without any oncological treatment. If no tumor residue was seen on the MRI one year after the end of the oncological treatments, the patient was then considered to be in complete remission

In order to take into account the various confounding factors, we performed a univariate analysis. Continuous variables were investigated using Student’s and Wilcoxon’s tests. Qualitative variables were studied with Fisher’s test or Chi-square. The significance level was set arbitrarily at 5%, i.e., an expected alpha probability value such as *p* < 0.05, with a 95% confidence interval (95% CI). Variables from the univariate analysis with a *p*-value < 0.1 were then included in the multivariate analysis. We performed a logistic regression to study the different factors that may interfere with the risk of relapse. These values were expressed in an odd ratio (OR) with a 95% confidence interval. We used R statistics software version 3.5.2.

## 3. Results

### 3.1. Population Characteristics

#### 3.1.1. Flow Chart

We included 124 patients (Figure 1).

#### 3.1.2. Comparison of the Population in the Two Groups

Median age at diagnosis was 4.5 years (0–17.6). Only 94 patients (76%) were referred to an endocrinologist.

Seventeen patients received GHRT (13.7%), with a median dose of 33 µg/kg/day (14–48) and with a median duration of 2.2 years (0.3–8). Population characteristics are described in Table 1.

Biopsy alone was not considered as a treatment.

Twenty patients had surgery only, without radiotherapy or chemotherapy: 19 in the group without GHRT and 1 in the group with GHRT. Two patients had only received first-line radiotherapy; they were in the group without GHRT. The median total dose of radiotherapy was 50.4 Grays (25.7–56). Thirty-one patients were only observed, without any treatment (30 patients in the group without GHRT and 1 patient in the group with GHRT).

Nine patients underwent total tumor resection (8.4%). Among them, there had been no deaths, and only one relapse (*p* = 0.08). All were in the group without GHRT. The other patients only benefited from partial excision (26 patients in the group without GHRT [24.3%] and 6 patients in the group with GHRT [35.3%]).

Moreover, none of the patients who could benefit from total excision had received adjuvant chemotherapy. Chemotherapy was therefore administered if the tumor was not operable or when the resection was incomplete. A total of 28 patients received a biopsy, 41 patients underwent complete or incomplete resection, and 55 patients therefore did not have an anatomo-pathological diagnosis. Among them, 42 patients were carriers of NF1. In the case of NF1, the diagnosis of OPG is based on its characteristic appearance of glioma on MRI and does not require histological proof [25]. Thirteen patients did not have NF1 and did not have an anatomo-pathological confirmation because the magnetic resonance imaging (MRI) was very suggestive of an OPG, and the biopsy was too risky for their location. For one of them, the biopsy had not been done because it was a therapeutic emergency to start chemotherapy.

There was no significant difference between the two groups, apart from two characteristics: number of patients with pilocytic astrocytoma was significantly lower in the group receiving GHRT (70.9% versus 88.8%, *p* = 0.02); number of patients treated with chemotherapy was higher in the group receiving GHRT (88.2% versus 49.5%, *p* = 0.003).

The median duration of follow-up since the end of oncological treatment was 8 years (4–211 months). On average, patients substituted with GH were followed longer after the end of their treatment than non-substituted patients. The average duration of oncological follow-up was 12.6 years (±33 months) in the group with GHRT, versus 7.8 years (±53.5 months) for children without GHRT (*p* < 0.001). The average time between the end of oncological treatment and the start of GHRT was 67.4 months, or 5.6 years (±31.3 months). All the characteristics of patients supplemented with GH are described in Table 2, and in Table 3 for patients not supplemented.

### 3.2. Primary Endpoint: Relapse

Nine relapses were observed out of the 17 patients receiving GHRT (52.9%), and 39 relapses were observed out of 107 without GHRT (36.4%) (*p* = 0.3). The median time to relapse in children with GHRT was 31 months (13–59) since the end of the last oncological treatment (*p* = 0.27), while it was 34 months (13–83) for patients without GHRT. The relapse-free survival rate over the entire duration of the study was 60.7% in the group without GHRT and 41.2% in the group with GHRT; the difference between the two groups was not significant (*p* = 0.13) (Figure 2).

Relapse was not significantly associated with age at diagnosis, gender, presence of NF1, or tumor type (Table 4). All of the analyses concerning the risk factors for relapse were carried out in univariate, then the risk of relapse was studied in multivariate for the variables with a *p*-value < 0.1 (resection and chemotherapy).

Only chemotherapy was associated with a higher risk of relapse in multivariate analysis (*p* = 0.043).

### 3.3. Second Cancer, Death and Diabetes

#### 3.3.1. Second Cancer Rate

Two patients (1.7%) developed a second cancer; both were carriers of NF1, and none of them had received GHRT. One patient had developed a mediastinal T-cell-lymphoma 4 years after treatment (the only treatment of his glioma was a complete surgical removal) and the second a cervical hamartoma 1.5 years after treatment. This patient had received only chemotherapy. The hamartoma is therefore probably linked to NF1.

#### 3.3.2. Death Rate

Five patients died (4.7%) during the study period: two due to tumor progression and three of treatment-related complications. None of them had received GHRT.

#### 3.3.3. Diabetes Rate

None of our patients developed type 2 diabetes on GHRT.

### 3.4. Particularities in GH Substituted Patients

Of all the patients substituted with GH, three patients (17,6%) presented with a relapse while undergoing treatment with GH (No. 47, 77, 91). GHRT was initiated, for two of them, after the first relapse, and after a one-year remission following this relapse (No. 47 and 108). Patient 108 had a second relapse while on GHRT. Patient 47 also relapsed on GHRT; this was his third relapse.

One patient (No. 91) relapsed her disease while receiving GHRT. This 7-month-old girl suffered from an optochiasmatic BRAF V600E-mutated ganglioglioma with a second localization in the 4th ventricle responsible for Russell’s syndrome and blindness at diagnosis. She was initially treated with chemotherapy (vincristine carboplatin). She then had a tumor recurrence 2.5 years later, treated with vinblastine and then with temozolomide, with partial surgery thereafter. GH replacement therapy was started 5 years after the end of any oncological treatment, since the tumor was stable. Nine months after starting GHRT, the patient again experienced tumor progression (Figure 3). The tumor then spontaneously shrank in size when GHRT was definitively stopped. She did not need tumor treatment for 5 years, then relapsed again. She still has sequelae hemiplegia, blindness, and was operated on for a fracture of the right femur due to a priori multifactorial osteoporosis (GH deficiency with contraindication to the resumption of supplementation, gonadotropic deficiency, and immobility).

The second patient (No. 47) had presented with two tumor recurrences before starting GHRT. Treatment with GH started 4.7 years after the last oncological treatment. A third relapse occurred six months after initiating treatment with GH.

Finally, a last patient (No. 77) was initially treated with chemotherapy (vincristine carboplatin) for a pilocytic astrocytoma of the third ventricle. Six months after starting GHRT, she had a first relapse. She was then treated by chemotherapy (vinblastine), subtotal resection, and then by proton beam therapy. She then developed gonadotropic, corticotropic, and somatotropic insufficiency. GHRT was subsequently resumed; she then had a second tumor recurrence 24 months after the start of the substitution. GHRT has been restarted for a third time at a distance from oncological treatments; it is still in progress at the present time, and the patient is still in remission 5 years later.

## 4. Discussion

Our study did not show a significant difference in the risk of relapse depending on whether patients treated for midline LGG received GHRT or not. Although there appears to be a trend towards an increased risk of recurrence in the GHRT group, the difference was not significant. Of course these results may be due to a lack of study power related to the relatively small number of patients who received GHRT. The retrospective nature of the study, the small number of patients, as well as a significant difference in patients between the two groups (17 patients with GHRT versus 107 patients without GHRT) represent obvious weaknesses in our study. These results should be confirmed by further studies. However, our cohort was relatively large (124 patients), and the patients were followed over a long period (median follow-up time: 8 years.).

The span from 1998 to 2016 is a wide timeframe, and there was variation in practice over this timespan concerning the use of growth hormone (GH). Before 1985, the treatment of GH deficiency consisted of replacing the subject’s GH by injecting GH extracted from cadaveric human pituitary glands. This was reserved for severe GHD. As early as 1985, GH could be synthesized thanks to molecular biology. This is the case for all patients in our study. The treatment is still very expensive, so the indications for treatment are still limited and also depend on an economic factor [24].

During the follow-up of the treatment with GH, the clinical and biological tolerance is evaluated. The World Health Organization (WHO, Geneva, Switzerland) Expert Committee on Biological Standardization (ECBS) has recognized the need for an international standard for IGF-1 for the calibration of immunoassays and for the control of content of therapeutic products [26]. Thanks to these other standardized dosages, the treatment with GH, which consisted solely of adapting the dose to the weight of the child, was able to evolve. The initial dose still depends on the indication: in the event of GH deficiency, 25 to 35 micrograms/kg/d, then the dose is adapted to clinical tolerance, growth rate, and IGF1 level. IGF1 should be maintained within the upper half of physiological values without exceeding them (<−2SD) [5,7].

The dose of GH and the time of introduction of GHRT (beyond 12 months after the end of the anticancer treatment) were consistent with the recommendations. There were no secondary malignancies following GHRT in our study, consistent with the reassuring data in the literature [27]. The median follow-up was longer in the GHRT group (12.6 years versus 7.8 years) (*p* < 0.001). This is probably explained by the fact that it is recommended to follow children for a longer time in case of GHRT for fear of a potential relapse [5,7,14].

In our study, only 14% of the children were supplemented with GH. However, according to the literature, GHD concerns approximately 40% of children treated for a brain tumor, and even more so in the case of a midline tumor [4]. This low percentage of children substituted in GH can be explained by two elements. First, only 76% of the children had benefited from endocrine monitoring in our study. However, this follow-up is recommended for all children with a tumor located in the pituitary region [8,10]. It is possible that some GH deficiencies were not seen in some patients due to a lack of endocrine monitoring. In fact, all of the children in the cohort were monitored for height and weight growth, but the search for growth hormone deficiency was not carried out systematically (only in the event of loss of stature or during a consultation with an endocrinologist). Children in whom growth retardation was most evident were probably substituted more often with GH. Furthermore, a number of parents and practitioners were probably reluctant towards growth hormone substitution for fear of relapse. We do not have clear figures on this.

The higher percentage of relapse in the group substituted with GH is probably also linked to the fact that patients who received GHRT had a more serious tumor than the others and were at greater risk of relapse. Indeed, there was a higher proportion of patients with three characteristics known in the literature to confer a better prognosis in the group without GHRT: the presence of NF1, pilocytic astrocytoma, and complete resection [1,28,29]. In addition, patients in the group without GHRT had received chemotherapy less often, which is used in the tumors most at risk of relapse [28]. The two groups were therefore not completely homogeneous with regard to these risk factors. These elements could explain the lower relapse rate in the group not supplemented with GH.

It should be noted that no patient in our study had benefited from a targeted therapy (MEK inhibitor or BRAF inhibitor), because these drugs were not available in France for the older patients, or not indicated. BRAF alterations (V600E mutations or BRAF-fusions) are frequently observed in LGGs. To date, targeted therapy is a promising option for pediatric gliomas harboring BRAF alterations [30]. Since these targeted therapies are very recent, we do not know yet precisely the duration of treatment (often 18 to 24 months, depending on the studies) [31,32]. Changing practices related to the prescription of MEK and BRAF inhibitors may lead to a review of the time required before GH is prescribed. This will need to be assessed by further studies.

As described before, one child particularly caught our attention: Patient 91 had relapsed only 9 months after initiation of GHRT, while the tumor had remained stable for 5 years before. This was his second relapse. When GHRT was stopped, the tumor spontaneously shrank in size, without oncological treatment. This patient then relapsed 5 years later, without any new GHRT. The association of these two events, namely the relapse under GHRT and the spontaneous tumor reduction on stopping GHRT, may turn out to be quite accidental, but they may legitimately raise the question of the safety of the GHRT. In addition, she suffered a fracture of the femur due to a priori multifactorial osteoporosis, which could possibly have been avoided if the resumption of the GHRT had been authorized.

Patient 47 also relapsed under GHRT, but this was then his third relapse. However, we know that patients who have already had a relapse are more likely to relapse in the future, independently of GHRT [1].

A minority of our patients (6%) received first-line radiotherapy (alone or combined with another treatment), and two of them had even received only radiotherapy. These are patients diagnosed with LGG in the 2000s, when the use of radiotherapy in LGG was common. Patients with a more recent diagnosis are treated with chemotherapy and/or surgery as the first line. In the study by Journy et al. studying the risk factors for relapse of all brain tumors after radiotherapy, relapse was not more frequent in the case of GHRT. However, the number of patients with glioma treated with GH was too small to analyze in this subgroup [33].

To our knowledge, to date, no study has specifically studied the risk of tumor recurrence in the event of treatment with GH in children treated for a low-grade glial tumor, which remains the primary etiology of brain tumors in children. Most of the studies looked at all types of brain tumors [2,7,10,11,15].

Despite the theoretical arguments regarding the pathophysiology of GH on carcinogenesis [34], there is no evidence of an increased risk of tumor recurrence after GH substitution.

A recent review encompassing safety data from the GH registries of various pharmaceutical companies between 1988 and 2016 showed no evidence of an increased risk of new malignancy, leukemia, brain tumor, or recurrence of malignant brain tumor in children treated with GH without any other risk factors [11]. In contrast, an increased risk of secondary cancer has been shown in patients irradiated for tumor of the central nervous system [11]. There may be an increased risk of type 2 diabetes in patients treated with GH, but this seems mostly confined to patients who have pre-existing risk factors for diabetes [13,35]; none of our patients were affected. The authors of this study conclude that patients with risk factors for cancer or type 2 diabetes, if they need to be treated, should be followed closely [12]. The published data are therefore rather reassuring in terms of the long-term safety profile of GH treatment.

The safety of GHRT after treatment of craniopharyngioma or even medulloblastoma has already been demonstrated. According to some studies, GHRT even reduces the risk of tumor recurrence [36,37]. A 2017 meta-analysis of 3487 patients with craniopharyngioma showed that the recurrence rate of the latter was significantly lower in children supplemented with growth hormone (10.9%; 95% CI: 9.8–12.1%) compared to children without supplementation (35.2%; 95% CI: 23.1–49.6%) [38].

Two major meta-analyses found a reduced risk of relapse in the event of GHRT [12,13]. In that of Shen L et al., progression or recurrence of intracranial tumors was not associated with GHRT (RR 0.48, 95% CI 0.39–0.56). In the subgroup analysis, the risk of recurrence and progression was decreased when patients received GHRT for craniopharyngioma, medulloblastoma, astrocytoma, or glioma but not for pituitary adenomas and ependymomas [12]. The meta-analysis by Zhi-Feng Wang et al., covering all brain tumors, found a tumor recurrence rate of 21.0% in children supplemented with GH and 44.3% in the GH-untreated group [13]. The pooled RR for recidivism was 0.470 (95% CI 0.372–0.593; *p* = 0.000). In the astrocytoma subgroup, the RR was 0.515 (95% CI 0.285–0.929, *p* = 0.028).

No studies have focused specifically on low-grade midline gliomas in GHRT. Only a few cohorts have been able to perform subgroup analyzes for astrocytomas based on their location [10,16,39]. Three cohorts of children supplemented with GH and suffering from various brain tumors did not find an increased risk of tumor recurrence in the subgroup of astrocytomas. Among the large cohort of Darendeliler et al., 400 patients were treated for a glial tumor. Among them, 39 presented a tumor recurrence (9.7%) [39]. The disease-free survival rate in these patients was 69% over 9.1 years of follow-up and was similar to literature data for children not supplemented with GHRT. Patients who relapsed were 1.5 years younger at diagnosis (*p* = 0.021) and were less than 2.2 years old at the start of GH treatment (*p* < 0.001) [39]. The author concludes, however, that prolonged follow-up for the detection of recurrences and secondary cancers remains essential. In the study by Swerdlow et al. [16], the risk of recurrence was lower for patients on GHRT, particularly for astrocytomas, with a relative risk of 0.5 (95% CI: 0.3–0.9). In the cohort of Sklar et al. [10], there was no increased risk of tumor recurrence in the event of treatment with GH (RR 0.98, CI 0.35–2.75, *p* = 0.96) in the 68 patients who had an astrocytoma.

We have summarized the GHRT in pediatric gliomas in Table 5. Our results are therefore in line with the various studies carried out in the past.

## 5. Conclusions

Our study did not find a significant difference in the rate of LGG tumor recurrence between children supplemented with GH and patients without supplementation. There were no deaths and no second cancers directly related to growth hormone supplementation.

Growth hormone supplementation therefore seems to be safe in these patients. It is nevertheless necessary to ensure the follow-up of the supplemented patients.

These results are in line with the data from recent literature and constitute an additional argument to reassure practitioners and parents about the safety of this treatment.

## Figures and Tables

**Figure 1 cancers-15-00055-f001:**
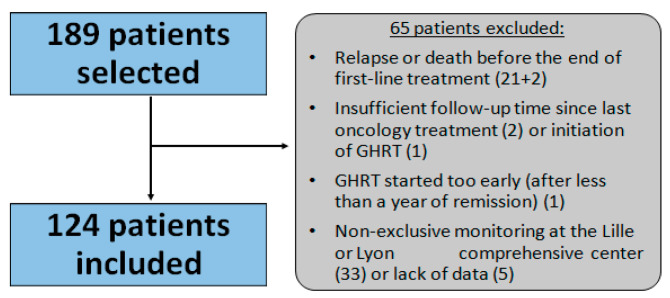
Flow chart depicting the selection of patients for the study. GHRT: growth hormone replacement therapy.

**Figure 2 cancers-15-00055-f002:**
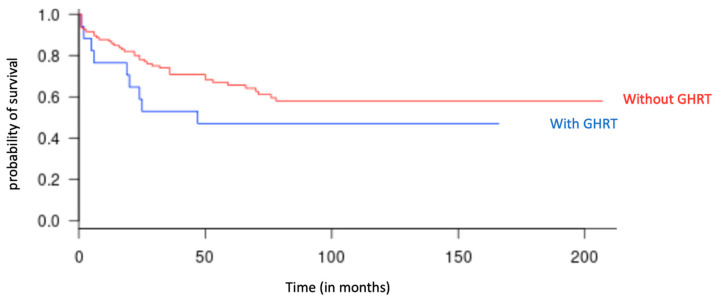
Relapse-free survival curve as a function of time (in months), in the groups with GHRT (in blue) and without GHRT (in red), over the entire duration of the study.

**Figure 3 cancers-15-00055-f003:**
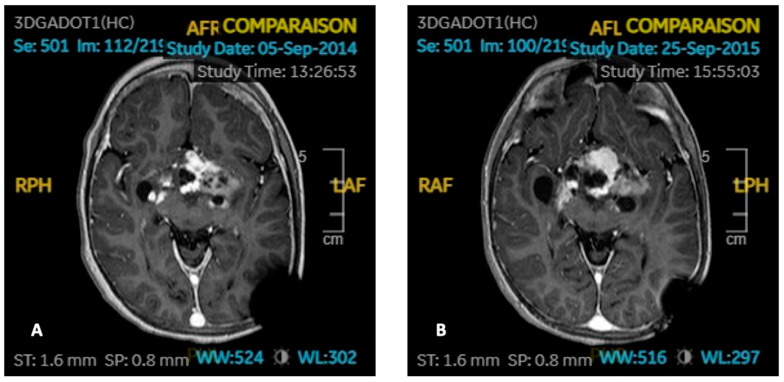
Imaging of Patient 91 with ganglioglioma (axial T1-weight with gadolinium). (**A**) Stable tumor residue before GHRT. (**B**) Progressive disease after 9 months of GHRT.

**Table 1 cancers-15-00055-t001:** Population characteristics.

	Group without GHRT	Group with GHRT	*p*-Value
Total population	*n* (%)	*n* (%)	
	107	17	
**Gender**			0.38
Female	60 (56.1%)	7 (41.2%)	
Male	47 (43.9%)	10 (58.8%)	
**NF1**	39 (36.5%)	3 (17.7%)	0.21
**Tumor type**			
Pilocytic astrocytoma (grade I)	95 (88.8%)	12 (70.6%)	0.02
Astrocytoma (grade II)	2 (1.9%)	3 (17.6%)	
Oligoastrocytoma (grade II)	3 (2.8%)	0 (0%)	
Oligodendroglioma (grade II)	3 (2.8%)	0 (0%)	
Ganglioglioma (grade II)	4 (3.7%)	2 (11.8%)	
**Biopsy (followed by chemo or RT or simple monitoring)**	22 (20.6%)	6 (35.3%)	0.21
**First-line treatments (can be combined)**			
Subtotal resection	26 (24.3%)	6 (35.3%)	0.37
Complete resection	9 (8.4%)	0 (0%)	1
Chemotherapy	53 (49.5%)	15 (88.2%)	0.003
Radiotherapy	7 (6.5%)	1 (5.9%)	0.99
**First-line treatments in detail**			
Surgery only	19 (17.8%)	1 (5.9%)	0.3
Surgery + RT	3 (2.8%)	0 (0%)	1
Surgery + chemo	14 (13.1%)	5 (29.4%)	0.14
Surgery + RT + chemo	1 (0.9%)	0 (0%)	1
Chemo only	37 (34.6%)	9 (52.9%)	0.18
RT only	2 (1.9%)	0 (0%)	1
Chemo + RT	1 (0.9%)	1 (5.9%)	0.26
**Simple monitoring**	30 (28%)	1 (5.9%)	0.07

NF1: Neurofibromatosis type 1; GHRT: growth hormone replacement therapy; RT: radiotherapy; Chemo: chemotherapy.

**Table 2 cancers-15-00055-t002:** Characteristics of patients treated with GH following a low-grade midline glial tumor between 1 January 1998 and 31 December 2016 at the Lille and Lyon CHUs.

Patients:	Age at Diagnosis (Year)	Sex	NF1	Histology	Localization	Surgery	Radio Therapy	Chemo Therapy	GH Onset Age (y)	GH Dose (μg/kg/d)	GH Duration (Months)	Relapse	Second Cancer	Death
No. 13	7.5	M	No	Grade II astrocytoma	Bilateral thalamus	B	-	-	15.4	48	47	No	No	No
No. 14	4.3	M	Yes	Pilocytic astrocytoma	OPG	-	-	Chemo	10.8	36	33	Yes ^#^ before GH	No	No
No. 15	2	F	No	Pilocytic astrocytoma	OPG	B	-	Chemo	10.9	38	20	Yes ^#^ before GH	No	No
No. 21	8.1	M	No	Grade II astrocytoma	Left thalamus	B	-	Chemo	12.5	32	12	No	No	No
No. 23	0.3	F	No	Grade II astrocytoma	Chiasmatoventricular	B	-	Chemo	9.5	22	*	No	No	No
No. 24	7	M	No	Pilocytic astrocytoma	Chiasmatoventricular	B	-	Chemo	13.5	29	40	Yes ^#^ before GH	No	No
No. 25	1.1	F	No	Pilocytic astrocytoma	Chiasmatoventricular	SR	-	Chemo	13.2	33	39	Yes ^#^ before GH	No	No
No. 27	3	M	No	Pilocytic astrocytoma	Chiasmatoventricular	SR	-	Chemo	10.75	37	54	Yes ^#^ before GH	No	No
No. 41	3.2	M	No	Pilocytic astrocytoma	OPG	-	-	Chemo	13.7	37	*	No	No	No
No. 43	6.6	M	No	Pilocytic astrocytoma	Chiasmatoventricular and V3	SR	-	Chemo	14.3	23	14	Yes ^#^ before GH	No	No
No. 45	1.3	F	Yes	Pilocytic astrocytoma	OPG	-	-	Chemo	10.5	41	*	No	No	No
No. 47	1.1	M	No	Pilocytic astrocytoma	OPG	SR	-	Chemo	13.75	36	7	Yes under GH	No	No
No. 51	0.7	F	No	Pilocytic astrocytoma	Right optic strip, diencephalon	SR	-	Chemo	10.2	33	*	No	No	No
No. 77	10.1	F	No	Pilocytic astrocytoma	OPG and V3	B	-	Chemo	12.8	35	6	Yes under GH	No	No
No. 81	4	M	Yes	Pilocytic astrocytoma	OPG	-	RT	Chemo	8.8	35	96	No	No	No
No. 83	13	M	No	Ganglioglioma	Suprasellar tumor and lateral ventricle	SR	-	-	20	15	24	No	No	No
No. 91	0.6	F	No	Ganglioglioma	OPG	-	-	Chemo	8.6	36	9	Yes under GH	No	No

M: male, F: feminine, NF1: Neurofibromatosis type 1, PA: Pilocytic Astrocytoma, OPG: Optic Pathway Glioma, V3: third ventricle, CR: complete resection, SR: Subtotal resection, B: Biopsy, Chemo: Chemotherapy, RT: Radiotherapy, * GH not interrupted at the time of data collection. ^#^ first relapse, then new remission before starting GH, but no relapse on GH.

**Table 3 cancers-15-00055-t003:** Characteristics of patients without GH following a low-grade midline glial tumor between 1 January 1998 and 31 December 2016 at the Lille and Lyon CHUs.

No.	Sex	NF1	Tumor	Localization	Surgery	RT	Chemo	Relapse	Death
1	M	-	PA	Thalamo-peduncular	SR	RT	-	-	-
2	F	-	PA	Chiasmato-ventricular	SR	-	Chemo	Relapse	-
3	M	-	PA	OPG	B	-	Chemo	-	-
4	F	-	PA	OPG	SR	-	Chemo	Relapse	-
5	F	NF1	PA	OPG	-	-	-	-	-
6	F	-	PA	Chiasmato-ventricular	SR	RT	Chemo	Relapse	-
7	M	NF1	PA	OPG	-	-	-	-	-
8	F	NF1	PA	OPG	-	-	-	-	-
9	F	-	PA	OPG	B	-	Chemo	-	-
10	F	-	PA	Thalamo-peduncular	SR	-	Chemo	Relapse	-
11	F	-	PA	Suprasellar tumor	-	-	Chemo	Relapse	-
12	M	-	PA	Chiasmato-ventricular	SR	-	Chemo	-	-
16	F	NF1	PA	OPG	-	-	-	-	-
17	M	-	Astrocytoma grade II	Right thalamus	SR	-	-	-	-
18	M	-	PA	Thalamo-peduncular	B	-	-	Relapse	-
19	M	-	Oligoastrocytoma	Left thalamus	SR	-	Chemo	-	-
20	M	-	PA	Right thalamus	CR	-	-	-	-
22	F	NF1	PA	OPG	-	-	-	-	-
26	M	NF1	PA	Right optic nerve	-	-	-	-	-
28	M	-	Oligodendroglioma	Thalamus and left lateral ventricle	CR	-	-	Relapse	-
29	F	-	PA	Chiasmato-ventricular	B	RT	-	-	-
30	F	-	Oligoastrocytoma	Right optic nerve	CR	-	-	-	-
31	F	NF1	PA	Right optic nerve	-	-	-	-	-
32	M	-	PA	Chiasmato-ventricular	SR	-	Chemo	Relapse	-
33	M	-	PA	Right temporo thalamo peduncular	SR	-	Chemo	-	Death
34	M	-	Oligodendroglioma	Right thalamus	B	RT	-	-	-
35	M	-	PA	OPG	B	-	Chemo	-	-
36	M	-	PA	Left optic nerve	CR	-	-	-	-
37	F	-	PA	Chiasmato-ventricular	SR	RT	-	Relapse	-
38	M	NF1	PA	OPG	-	-	-	-	-
39	M	-	PA	OPG infiltrating basal ganglia	B	-	Chemo	Relapse	-
40	F	NF1	PA	OPG	-	-	Chemo	Relapse	-
42	F	NF1	PA	OPG	-	-	-	Relapse	-
44	M	-	PA	Chiasmato-ventricular	B	-	Chemo	Relapse	Death
46	M	NF1	PA	Left optic nerve	CR	-	-	-	-
48	M	NF1	PA	OPG	-	-	-	-	-
49	F	-	PA	V3	CR	-	-	-	-
50	F	-	Oligodendroglioma	Right thalamus	B	-	-	-	-
52	F	NF1	PA	OPG	-	-	-	-	-
53	F	NF1	PA	OPG	-	-	Chemo	Relapse	-
54	F	NF1	PA	OPG	-	-	Chemo	Relapse	-
55	F	NF1	PA	OPG	-	-	Chemo	Relapse	-
56	F	-	PA	Left thalamus	CR	-	Chemo	-	-
57	M	-	Astrocytoma grade II	OPG and diencephalon	B	-	Chemo	-	-
58	F	-	PA	OPG	SR	RT	-	-	-
59	F	-	PA	OPG	-	-	Chemo	-	-
60	F	-	Oligoastrocytoma	Right V3 and thalamus	B	-	Chemo	Relapse	-
61	M	-	Ganglioglioma	Right occipital ventricular junction tumor	SR	-	-	Relapse	-
62	M	-	PA	Suprasellar tumor	SR	-	Chemo	Relapse	-
63	F	-	PA	Missing data	B	-	Chemo	-	-
64	F	-	PA	OPG	SR	-	Chemo	Relapse	Death
65	M	NF1	PA	Left optic nerve	-	-	-	-	-
66	F	NF1	PA	OPG	-	-	-	-	-
67	F	-	PA	V3	CR	-	-	-	-
68	M	-	PA	OPG	SR	-	Chemo	Relapse	-
69	M	-	PA	OPG	SR	-	Chemo	Relapse	-
70	F	-	PA	Suprasellar tumor	SR	-	Chemo	Relapse	-
71	F	-	PA	OPG	-	-	Chemo	-	-
72	M	-	PA	Right thalamus	SR	-	-	-	-
73	M	NF1	PA	OPG	-	-	-	-	-
74	F	-	PA	Suprasellar tumor	SR	-	-	Relapse	-
75	M	-	PA	OPG and V3	SR	-	-	Relapse	-
76	F	NF1	PA	OPG	-	-	-	-	-
78	F	-	PA	Quadruple blade	SR	-	-	-	-
79	M	NF1	PA	OPG	-	-	Chemo	-	Death
80	F	-	PA	Thalamo-peduncular	B	-	Chemo	-	-
82	F	-	Ganglioglioma	V3	SR	-	-	-	-
84	F	-	PA	Retrochiasmatic lesion	-	-	-	Relapse	-
85	M	-	Ganglioglioma	V3	CR	-	-	-	-
86	F	-	PA	Suprasellar tumor	B	-	-	-	-
87	F	NF1	PA	OPG	-	-	-	-	-
88	F	NF1	PA	OPG	-	-	-	Relapse	-
89	F	NF1	PA	OPG	-	-	-	-	-
90	F	NF1	PA	OPG	-	-	-	-	-
92	F	-	PA	OPG	B	-	Chemo	-	-
93	F	-	PA	OPG	-	-	Chemo	Relapse	-
94	M	-	PA	OPG	SR	-	-	Relapse	-
95	F	-	PA	OPG	B	-	Chemo	-	-
96	F	NF1	PA	OPG and V3	-	-	-	Relapse	-
97	F	-	PA	OPG	B	-	Chemo	-	-
98	M	-	PA	OPG	-	-	Chemo	-	-
99	F	-	PA	OPG	-	-	Chemo	-	-
100	F	-	PA	OPG	-	-	Chemo	-	-
101	F	NF1	PA	OPG	-	-	-	Relapse	-
102	M	NF1	PA	OPG	-	-	-	-	Death
103	M	-	PA	Thalamo-peduncular	B	-	Chemo	-	-
104	M	-	PA	Right thalamus and V3	B	-	Chemo	-	-
105	M	-	Ganglioglioma	Hypothalamus	-	-	-	Relapse	-
106	F	-	PA	Left thalamus	B	-	Chemo	Relapse	-
107	M	NF1	PA	OPG	-	-	-	Relapse	-
108	M	NF1	PA	OPG	-	-	-	-	-
109	M	NF1	PA	OPG	-	-	Chemo	-	-
110	M	NF1	PA	OPG	-	-	Chemo	-	-
111	M	-	PA	OPG	-	-	Chemo	Relapse	-
112	F	NF1	PA	OPG	-	-	Chemo	-	-
113	M	NF1	PA	OPG	-	-	Chemo	Relapse	-
114	F	NF1	PA	OPG	-	-	Chemo	-	-
115	F	NF1	PA	OPG	-	-	Chemo	Relapse	-
116	F	NF1	PA	OPG	-	-	Chemo	-	-
117	F	NF1	PA	OPG	-	-	Chemo	-	-
118	M	-	PA	OPG	-	RT	Chemo	-	-
119	M	-	PA	OPG	B	-	Chemo	Relapse	-
120	M	-	PA	OPG	SR	-	-	-	-
121	F	-	PA	Right capsulo-thalamic	B	-	-	-	-
122	M	-	PA	OPG	-	-	-	-	-
123	F	NF1	PA	OPG	-	-	-	Relapse	-
124	F	-	PA	Suprasellar and V3	SR	-	Chemo	-	-

M: male; F: female; NF1: neurofibromatosis type 1; PA: pilocytic astrocytoma; OPG: optic pathway glioma; V3: third ventricle; CR: complete resection; SR: subtotal resection; B: biopsy; Chemo: chemotherapy; RT: radiotherapy.

**Table 4 cancers-15-00055-t004:** The impact of different factors on relapse in patients treated for midline glioma in Lille and Lyon between 1 January 1998 and 31 December, 2016 (univariable analysis).

	Patients who Did Not Relapse		Patients who Did Relapse		
	*n* = 76	%	*n* = 48	%	*p*-Value
**Age at diagnosis**					0.96
<12 months	8	10.5	6	12.5	
>12 months	68	89.5	42	87.5	
**Gender**					0.83
Female	40	52.6	27	56.3	
Male	36	47.4	21	43.7	
**NF1**					0.28
without NF1	47	61.8	35	72.9	
with NF1	29	38.2	13	27.1	
**Tumor type**					0.49
Pilocytic astrocytoma (grade I)	64	84.2	43	89.6	
Astrocytoma (grade II)	5	6.6	0	-	
Oligoastrocytoma (grade II)	2	2.6	1	2.1	
Oligodendroglioma (grade II)	2	2.6	1	2.1	
Ganglioglioma (grade II)	3	4	3	6.2	
**First line chemotherapy**					0.013
Chemotherapy	35	46	33	68.7	
No chemotherapy	41	54	15	31.3	
**First line radiotherapy**					0.47
Radiotherapy	6	7.9	2	4.2	
No radiotherapy	70	92.1	46	95.8	
**First line type of surgery**					0.088
Complete resection	8	10.5	1	2. 1	
Subtotal resection	13	17. 1	19	86.4	
**GHRT**					0.28
no GHRT	68	89.5	39	81.3	
with GHRT	8	10.5	9	18.7	

NF1: neurofibromatosis type 1; GHRT: growth hormone replacement therapy.

**Table 5 cancers-15-00055-t005:** Main articles in the literature concerning growth hormone supplementation in children with brain tumors.

References	Study Groups	Recurrence	Authors Conclusions
Karavitaki et al., 2006 [40]	Craniopharyngioma: 32 patients with GHRT (but 11 started during adult life), 53 without GHRT	4 patients treated with GH and 22 non-GH treated ones developed tumour recurrence (*p* = 0.06; RR = 0.309)	GH replacement does not increase the risk of recurrence in patients with craniopharyngioma
Rohrer et al., 2010 [41]	108 craniopharyngioma, medulloblastoma, and ependymoma patients	13/44 GH-treated and 28/59 non-GH-treated children relapsed	No increased risk of recurrence under GHRT
Alotaibi et al., 2017 [38]	Craniopharyngioma: 3436 pediatric patients were treated with GHRT after surgery (GHRT duration ranged between 1.9 and 6.4 years), and 51 were not.	The recurrence rate of the latter was significantly lower in children supplemented with GH (10.9%; 95% CI: 9.80–12.1%) compared to children without supplementation (35.2%; 95% CI: 23.1–49.6%)	This meta-analysis demonstrated a lower recurrence rate of craniopharyngioma among children treated with GHRT than those who were not.
Shen et al., 2015 [12]	All brain tumors, meta-analysis of 15 studies, 2232 patients with GHRT and 3606 without GHRT	RR of recurrence: 0.44, (95% CI = 0.34 to 0.54; *p* = 0.680) In the subgroup analysis, the risks of recurrence were decreased for craniopharyngioma, medulloblastoma, astrocytoma, glioma, but not for pituitary adenomas and ependymomas	Recurrence of intracranial tumors was not associated with GHRT
Zhi-Feng Wang et al., 2014 [13]	All brain tumors, meta-analysis of 10 studies	Tumor recurrence rate of 21.0% in children with GHRT and 44.3% without GHRT. RR for recidivism: 0.470 (95% CI 0.372–0.593; *p* = 0.000). In the astrocytomas subgroup, the RR was 0.515 (95% CI 0.285–0.929, *p* = 0.028)	No increased risk of recurrence under GHRT
Darendeliler et al., 2006 [39]	400 patients were treated for a glial tumor with GHRT	39 presented a tumor recurrence (9.7%) The disease-free survival rate in these patients was 69% over 9.1 years of follow-up, similar to literature data for children not supplemented with GHRT.	Recurrence of glial tumors was not associated with GHRT. Prolonged follow-up for the detection of recurrences and secondary cancers remains essential.
Swerdlow et al., 2000 [16]	All brain tumors after radiotherapy, 180 children with GHRT and 891 children without GHRT.	Thirty-five first recurrences occurred in the GH-treated children and 434 in the untreated children. RR of first recurrence for all brain tumors: 0.6; 95% CI 0.4–0.9). RR for astrocytomas: 0.5 (95% CI: 0.3–0.9).	Risk of recurrence was lower for patients on GHRT, particularly for astrocytomas

## Data Availability

The data presented in this study are available in this article (Table 2 and Table 3).

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
