# Peer review of "Growth Hormone Replacement Therapy Seems to Be Safe in Children with Low-Grade Midline Glioma: A Series of 124 Cases with Review of the Literature"

_cancers, 2022, doi:10.3390/cancers15010055_

Round 1

Reviewer 1 Report

Reviewer general comment: This study is important to review growth hormone use in pediatric patients with midline-located low-grade gliomas (LGG). This topic has the potential to inform practice. Although they had a relatively large number of children with midline LGGs (n = 124), there were only 17 that were supplemented with growth hormone. This leads to a small number of patients and therefore results of any statistical analyses will be exploratory and firm conclusions need to be cautioned against – Per power calculations, you will be able to detect somewhere around a 20% increase in relapse risk with current numbers although there will be likely selection bias as well for which patients received growth hormone supplementation which will be difficult to account for given the retrospective nature of the study and clinician judgment/decision making. That said, it is appreciated that the authors took steps to ensure some uniformity of data for the analyses including single tumor type and no GHRT before 1 year off therapy. I liked that the authors did a review of the literature on this important topic and I feel that this adds value to the manuscript. However, a supplemental table or chart would be useful to better summarize this data.  Overall, it is an important effort and manuscript but some improvements are needed to improve the quality and avoid overstepping the conclusions that can be made.  

Major

(1)    The conclusion of the abstract and manuscript is overreaching and should be softened. See a and below.

a.       Abstract Results: line 43 should say no “statistically” significant difference as one could argue that 52.9% vs 36.4% might be clinically significant.  Of course, numbers are limited in the GH cohort.

b.       Abstract Conclusion: I would qualify the initial sentence on lines 45-46 that “GHRT does not appear to lead to a statistically significant increase risk of relapse for pediatric midline LGG in our cohort.” The second sentence should have a qualifier “Although these results appear reassuring, future natural history or prospective studies should verify these findings.”

(2)    Review of the literature.

(3)    This study does not comment on the use of targeted therapies in LGGs which is becoming increasingly common. The relevance of this is that using MEK inhibitors +/- BRAF inhibitors is more difficult to know when to discontinue these anticancer agents and in turn how do we approach GH deficiency in these children that may be on prolonged or possibly indefinite anticancer therapy. This item should be commented on in the discussion. It should also be noted if any of your 124 patients were receiving targeted therapy (MEKi or BRAFi) at any time.

(4)    How many patients overall had GH deficiency in the non-GHRT cohort?

(5)    Methods:

a.       Can you define treatment for the readership?  Is a biopsy alone considered treatment, or did all patients in the non-GHRT receive chemotherapy or radiation? Did some patients have surgery alone without chemotherapy or radiation?

(6)    Results:

a.       Can you let the readers know if there were differences between the number of patients who were treated with surgery alone in the GHRT group vs the no GHRT group?

b.       Although not statistically significant, the readership is not clear what difference in relapse rate your study was powered to detect.  I think we need clarification here because of the smaller numbers in the GHRT group.

c.       Was there any molecular data for your cohort? This would be great to assess BRAF V600E vs KIAA1549-BRAF fused?

d.       Can you inform the audience what number of patients from your cohorts had clinical vs laboratory-confirmed GH deficiency vs which patients were considered GH intact?  How were these patients found to have GH deficiency?

Minor

(2)    Simple summary

a.       Is there any significance in having both “mitogenic and proliferative” – perhaps just choose one descriptor?

b.       Is it true that growth hormone deficiency is the most common endocrine sequelae in children with brain tumors? In https://doi.org/10.1093/neuonc/noq178 it appears that obesity is more common. I challenge the authors to note more citations for their claim or soften their language or clarify that it is the most common hypothalamic-pituitary disorder.

c.       The last sentence of the summary “constitutes an additional argument to reassure practitioner and parents…”  What are the other arguments, it seems that only relapse is discussed.  Perhaps “These results constitute important findings that contribute to the safety of growth hormone in this population.”

(3)    Abstract:

a.       Please review the grammar.

b.       Methods: line 38 median should be “midline”.

(4)    Introduction:

a.       The time frame in the introduction from Jan 1998 to Dec 2019 is different than in the introduction from 1998 to 2016.

(5)    Results:

a.       Line 173 – Please clarify the median time to relapse was 31 months for those with GHRT and 34 months for patients without GHRT.  Is this from the time of initial diagnosis? Or from the end of therapy?

b.       Please clarify that table 2 is the univariable data, is this correct?

(6)    Discussion:

a.       1998 to 2016 is a wide timeframe – I am sure there was variation in practice over this timespan concerning use of growth hormone.  Can the authors comment on this in their discussion?

b.       Many of your patients receive first-line radiotherapy.  This is no longer considered the standard of care in North America for LGG.  This should be commented upon in the discussion.

c.       Line 260 – relatively large (144 patients) – should be 124 patients.

d.       Line 273 – states “only 16% of our cohort had GHRT” but it is 14% as per your results section.

e.       Line 284-292 – the authors feel the need to justify the difference in percentages between relapse in the GHRT group and the non-GHRT group, despite their conclusion stating this is not significant.  This is why it is important to be clear about your limitations regarding the statistical strengths/weaknesses of this paper and the exploratory nature.

f.        The sentence on line 323 “According to some studies, GHRT even reduces the risk of tumor recurrence.” Needs to be cited appropriately as it cannot stand alone.

g.       Please consider a supplemental table for your literature review to better summarize the GHRT in pediatric gliomas.

Author Response

Reviewer 1

Open Review

Comments and Suggestions for Authors

Reviewer general comment: This study is important to review growth hormone use in pediatric patients with midline-located low-grade gliomas (LGG). This topic has the potential to inform practice. Although they had a relatively large number of children with midline LGGs (n = 124), there were only 17 that were supplemented with growth hormone. This leads to a small number of patients and therefore results of any statistical analyses will be exploratory and firm conclusions need to be cautioned against – Per power calculations, you will be able to detect somewhere around a 20% increase in relapse risk with current numbers although there will be likely selection bias as well for which patients received growth hormone supplementation which will be difficult to account for given the retrospective nature of the study and clinician judgment/decision making. That said, it is appreciated that the authors took steps to ensure some uniformity of data for the analyses including single tumor type and no GHRT before 1 year off therapy. I liked that the authors did a review of the literature on this important topic and I feel that this adds value to the manuscript. However, a supplemental table or chart would be useful to better summarize this data.  Overall, it is an important effort and manuscript but some improvements are needed to improve the quality and avoid overstepping the conclusions that can be made.  

Major

(1)    The conclusion of the abstract and manuscript is overreaching and should be softened. See a and below.

  1. * Abstract Results: line 43 should say no “statistically” significant difference as one could argue that 52.9% vs 36.4% might be clinically significant.  Of course, numbers are limited in the GH cohort.

>> we have modified the text as requested:

“Relapse concerned 36.4% of patients without GHRT and 52.9% of patients with GHRT, however, the difference was not significant between the two groups”

  1. *Abstract Conclusion: I would qualify the initial sentence on lines 45-46 that “GHRT does not appear to lead to a statistically significant increase risk of relapse for pediatric midline LGG in our cohort.” The second sentence should have a qualifier “Although these results appear reassuring, future natural history or prospective studies should verify these findings.”

>> we have modified the text as requested:

“GHRT does not appear to lead to a statistically significant increase risk of relapse for pediatric midline low grade glioma in our cohort. Although these results appear reassuring, future natural history or prospective studies should verify these findings.”

(2)    Review of the literature.

(3)    This study does not comment on the use of targeted therapies in LGGs which is becoming increasingly common. The relevance of this is that using MEK inhibitors +/- BRAF inhibitors is more difficult to know when to discontinue these anticancer agents and in turn how do we approach GH deficiency in these children that may be on prolonged or possibly indefinite anticancer therapy. This item should be commented on in the discussion. It should also be noted if any of your 124 patients were receiving targeted therapy (MEKi or BRAFi) at any time.

Thank you very much for this very interesting comment. Indeed, we do not comment on the use of targeted therapies because the prescription of targeted therapies began after 2016 in France. None of our patients benefited from a targeted treatment (MEKi or BRAFi). Nevertheless, indeed, chemotherapy has a duration limited to I8 months in LGG but we do not know very well the ideal duration of targeted therapies (it is between I8 and 24 months depending on the studies). The time required before the introduction of a GHRT after a targeted therapy will have to be clarified by subsequent studies.

We have modified the text as follows:

“It should be noted that no patient in our study had benefited from a targeted therapy (MEK inhibitor or BRAF inhibitor) because these were not generalized in France until after 2016. Since these targeted therapies are very recent, we do not know not yet precisely the duration of treatment. It is 18 to 24 months depending on the studies.

(We have added 2 references to clarify this; Phase II open-label global study to evaluate the effect of dabrafenib in combination with trametinib in children and adolescent patients with BRAF V600 mutation positive Low Grade Glioma (LGG) or relapsed or refractory High Grade Glioma (HGG), and, Low Grade Glioma - MEKinhibitor TRIal vs Chemotherapy (PLGG - MEKTRIC). Changing practices related to the prescription of MEK and BRAF inhibitors may lead to a review of the time required before GH is prescribed. This will need to be assessed by further studies.”

 (4)    How many patients overall had GH deficiency in the non-GHRT cohort?

  • In our study, all patients were monitored for height and weight growth. The deficiency in growth hormone was only in the event of a break in the height curve, by an IGF1 assay and by a growth hormone stimulation test in case of doubt. We therefore do not have an IGF1 assay for all of the patients in the study.

  • We have modified the text as follows, line 152:
  • “In our study, all patients were monitored for height and weight growth. The deficiency in growth hormone was only in the event of a break in the height curve, by an IGF1 assay and by a growth hormone stimulation test in case of doubt.”

(5)    Methods:

*a.       Can you define treatment for the readership?  Is a biopsy alone considered treatment, or did all patients in the non-GHRT receive chemotherapy or radiation? Did some patients have surgery alone without chemotherapy or radiation?

Thank you very much for your remark. Indeed, our table was not clear, we corrected it in order to clarify that. Biopsy alone was not considered a treatment.

20 patients had only surgery, without radiotherapy or chemotherapy; 19 in the group without GHRT and one in the group with GHRT.

We have also changed the text as follows:

” Biopsy alone was not considered a treatment.

Twenty patients had only surgery, without radiotherapy or chemotherapy; 19 in the group without GHRT and one in the group with GHRT. 2 patients had only received first-line radiotherapy, they were in the group without GHRT. The median total dose of radiotherapy was 50.4 Grays (25.7 - 56). 31 patients had simple monitoring, without any treatment (30 patients in the group without GHRT and one patient in the group with GHRT).

Nine patients had benefited from total excision (8.4%). Among them, there had been no death, and only one relapse (p=0.08). They were all in the group without GHRT. The other patients could only benefit from partial excision (26 patients in the group without GHRT (24.3%) and 6 patients in the group with GHRT (35.3%).

Furthermore, none of the patients who could benefit from total excision had received adjuvant chemotherapy. Chemotherapy was therefore administered if the tumor was not operable, or when the resection was incomplete.”

 (6)    Results:

  1. Can you let the readers know if there were differences between the number of patients who were treated with surgery alone in the GHRT group vs the no GHRT group?

20 patients had only surgery, without radiotherapy or chemotherapy; 19 in the group without GHRT and one in the group with GHRT. The difference was not significative (p=0,3). We corrected the table.

  1. Although not statistically significant, the readership is not clear what difference in relapse rate your study was powered to detect.  I think we need clarification here because of the smaller numbers in the GHRT group.

Thank you for your remark;

The study was not weighted to detect a relapse rate,

This is a retrospective cohort study unweighted to detect a relapse rate. We therefore included all patients treated for LGG over a given period, regardless of the number of patients to be included.

  1. Was there any molecular data for your cohort? This would be great to assess BRAF V600E vs KIAA1549-BRAF fused?

Indeed, thank you very much for this comment. That would indeed have been very interesting; but we had no molecular data for this cohort.

At the time, between 1998 and 2016, these examinations were not done routinely.

We cannot reanalyze all the anatomo-pathological parts retrospectively.

  1. Can you inform the audience what number of patients from your cohorts had clinical vs laboratory-confirmed GH deficiency vs which patients were considered GH intact?  How were these patients found to have GH deficiency?
  • In our study, all patients were monitored for height and weight growth. The deficiency in growth hormone was only in the event of a break in the height curve, by an IGF1 assay and by a growth hormone stimulation test in case of doubt. We therefore do not have an IGF1 assay for all of the patients in the study.

  • We have modified the text as follows, line 152:
  • “In our study, all patients were monitored for height and weight growth. The deficiency in growth hormone was only in the event of a break in the height curve, by an IGF1 assay and by a growth hormone stimulation test in case of doubt.”

Minor

(2)    Simple summary

*a.       Is there any significance in having both “mitogenic and proliferative” – perhaps just choose one descriptor?

>> We chose “proliferative”.

*b.       Is it true that growth hormone deficiency is the most common endocrine sequelae in children with brain tumors? In https://doi.org/10.1093/neuonc/noq178 it appears that obesity is more common. I challenge the authors to note more citations for their claim or soften their language or clarify that it is the most common hypothalamic-pituitary disorder.

>> We have modified the text as requested

Growth hormone (GH) deficiency is the most common hypothalamic-pituitary disorder due to a brain tumor during childhood, whether it is related to the tumor itself or to the treatment received.

We added this reference.

*c.       The last sentence of the summary “constitutes an additional argument to reassure practitioner and parents…”  What are the other arguments, it seems that only relapse is discussed.  Perhaps “These results constitute important findings that contribute to the safety of growth hormone in this population.”

>> We have corrected the sentence by your suggestion, thank you very much for this comment.

(3)    Abstract:

  1. Please review the grammar.

*b.       Methods: line 38 median should be “midline”.

>> Sorry yes, thank you very much.

(4)    Introduction:

*a.       The time frame in the introduction from Jan 1998 to Dec 2019 is different than in the introduction from 1998 to 2016.

>> We included all patients diagnosed between 1998 and 2016, then we collected their data until 2019, in order to have at least 3 years of follow-up, hence this difference in date. We are therefore changing the date here to 2016 for clarity.

(5)    Results:

  1. Line 173 – Please clarify the median time to relapse was 31 months for those with GHRT and 34 months for patients without GHRT.  Is this from the time of initial diagnosis? Or from the end of therapy?

>> It was since the end of the last oncological treatment: we have corrected the sentence.

  1. Please clarify that table 2 is the univariable data, is this correct?

>> Yes, it is, we fixed that.

(6)    Discussion:

  1. 1998 to 2016 is a wide timeframe – I am sure there was variation in practice over this timespan concerning use of growth hormone.  Can the authors comment on this in their discussion?

>> Thank you for your very relevant comment. Indeed, 1998 to 2016 is a wide timeframe and  there was variation in practice over this timespan concerning use of growth hormone (GH).

Before 1985, the treatment of GH deficiency consisted of substituting the subject’s GH by injecting GH extracted from cadaveric humans pituitary glands. It was reserved for severe GHD. 

From 1985, GH could be synthesized thanks to molecular biology. The treatment is still very expensive, therefore the indication for treatment are still limited and also depend on an economic factor. 

For GH deficiency diagnosis, IGF1 is measured and IGFBP3 is measured in case of younger child too.  GH should be mesured after stimulation in France by hypoglycaemia (insulin tolerance test, ITT), by L dopa, arginine, glucagon, propranolol, clonidine or GHRH (Official Journal of January 1997). These stimulation tests can be used in combination. The decision threshold is set by order of the Official Journal of march 2003 which indicates that the diagnosis of GH deficiency must be proven by 2 tests, one simple and one combined stimulation test (Official Journal16 march 2003): if both of them show a result lower than 10 mUI/ml (3 micrograms/L), there is a severe GH deficiency. There is a partial GH deficiency if the results are between 10 and 20mUI/L (3 to 6,7micrograms/L). A single test with response >6,7 micrograms/L rules out the diagnosis of GHD Official Journal of January 1997). If there is an obvious acquired cause of GHD, only one stimulation test, with IGF1 dosage is necessary.

An international working group has establish a consensus to propose standardization of GH dosage on the 98/574 standard. The results should be expressed in micrograms/L (Official Journal 29 january 1997). During monitoring of GH treatment, clinical and biological tolerance are evaluated. The World Health Organization (WHO) Expert Committee on Biological Standardization (ECBS) has recognized the need for an International Standard for Insulin-like Growth Factor-1 (IGF-1) for the calibration of immunoassays and for the monitoring of the content of therapeutic products.IGF1 dosage has been standardized too on WHO standard 02/254 . The objective  of standardization is to obtain “correct and reproducible” biological dosage (The First International Standard For Insulin-like Growth Factor-1 (IGF-1) for immunoassay: preparation and calibration in an international collaborative study.

C Burns et al. Growth Horm IGF Res. 2009 Oct.; 

Clemmons, DR. consensus statement on the standardization and evaluation of GH and IGF assays.Clin Chem 57, 555-559 (2011).

Due to this other standardized dosages, GH treatment which consisted only of adapting the dose to  the child weight, was able to evolve. The initial dose still depends on the indication: in GH deficiency, 25 to 35 micrograms/kg/d, then, the dose is adapted to clinical tolerance (joint pain…), growth rate and IGF1 level. IGF1 can be maintained in the upper half of the physiological values without exceeding them. 

We have included some of these elements in the manuscript (methods and discussion).

  1. Many of your patients receive first-line radiotherapy.  This is no longer considered the standard of care in North America for LGG.  This should be commented upon in the discussion.

I had inadvertently reversed the lines in table 2, I'm really sorry, I corrected that.

Indeed, 8 of our patients (6%) received first-line radiotherapy (alone or combined with another treatment), and 2 of them had even received only radiotherapy. These are patients diagnosed with LGG in the 2000s. Patients diagnosed more recently are treated with first-line chemotherapy and/or surgery.

We have modified the text as follows:

“Height of our patients (6%) received first-line radiotherapy (alone or combined with another treatment), and 2 of them had even received only radiotherapy. These are patients diagnosed with LGG in the 2000s, when the use of radiotherapy in LGG was common. Patients with a more recent diagnosis are treated with chemotherapy and/or surgery as the first line.”

  1. Line 260 – relatively large (144 patients) – should be 124 patients.

Sorry for this error, we have corrected this, thank you very much.

  1. Line 273 – states “only 16% of our cohort had GHRT” but it is 14% as per your results section.

Sorry for this error, we have corrected this, thank you very much.

  1. Line 284-292 – the authors feel the need to justify the difference in percentages between relapse in the GHRT group and the non-GHRT group, despite their conclusion stating this is not significant.  This is why it is important to be clear about your limitations regarding the statistical strengths/weaknesses of this paper and the exploratory nature.

It is certain that our study had statistical weaknesses.

Our study probably involved too small a sample of patients, and an insufficient number of patients substituted with GH. It was indeed difficult to know the number of patients substituted in GH before starting the collection.

To increase the number of patients in the study, we could have collected data on even earlier years: but the collection would have been difficult to carry out because the medical records were only computerized in 1998 in Lille and in 2000 in Lyons. It would therefore have been necessary to include other centers to increase the statistical power.

The selection bias was limited in our study since we had included all the patients treated in Lille and Lyon over a specific period. This population sample was representative of the general population since there is only one oncology center for a large population pool in each of these 2 regions.

The measurement bias of the primary endpoint was very limited because each diagnosis of relapse was confirmed during a multidisciplinary consultation meeting (RCP). Each MRI was reread in the presence of neurosurgeons, oncologists and radiologists during this meeting in order to validate or not the reality of tumor recurrence. When investigating the collection of data, we relied on the decisions taken during these RCPs.

Nevertheless, this study remains a retrospective, exploratory study, with all the limitations that go with it.

We added this sentence in the manuscript:

The retrospective nature of the study, the small number of patients, as well as a significant difference in patients between the two groups (17 patients with GHRT versus 107 patients without GHRT) represent obvious weaknesses in our study. These results should be confirmed by further studies.

  1. The sentence on line 323 “According to some studies, GHRT even reduces the risk of tumor recurrence.” Needs to be cited appropriately as it cannot stand alone.

We have corrected this, thank you very much.

  1. Please consider a supplemental table for your literature review to better summarize the GHRT in pediatric gliomas.

Thank you for this suggestion. We added this table in the manuscript:

Table 4. Main articles in the literature concerning growth hormone supplementation in children with brain tumors.

Reviewer 2 Report

Puvilland et al provide a quantitative analysis of the effect of growth hormone replacement therapy (GHRT) on tumor relapse & prognosis of low-grade glioma patients with midline tumors. The authors very clearly explain the rationale behind the selection of tumor patients for the study (emphasizing the relationship between the location of the glioma & requirement of the growth hormone therapy). The authors compare all necessary factors including the tumor relapse, tumor prognosis, the type of glioma, the therapy administered to the patient, and NF1 mutational status, among others, and found that chances of tumor relapse were not affected among patients with or without GHRT. The manuscript was easy to understand, very well written and had all necessary data available for analyze the retrospective study. The manuscript should be accepted with certain concerns mentioned in the attached file.

Author Response

Reviewer 2

Comments and Suggestions for Authors

Authors present a bicentric retrospective study with 124 patients under the age of 18, diagnosed with a median low-grade glial tumor between 1998 and 2016. There were 17 patients who were supplemented with growth hormone (14%) and 107 patients in the unsupplemented group (86%). Relapse occurred in 65 patients (45.5%), 7 patients died (4.9%): no death  occurred in patients receiving GHRT.  There was no significant difference between the two groups since relapse concerned. Authors conclude in their observations that GHRT seems to be safe in children with midline low grade glioma, as no significant  risk of relapse has been demonstrated.

Low number of patients, retrospective character of the study as well as large discrepancy between the two groups (17 vs. 107 patients) are major drawbacks of the study, so no serious conclusion can be statistically drawn. However, this is a rare tumor and any clinical experience is welcome, however lot more clinical , neuroradiological and neuropathological data with thorough revision of the manuscript , i.e. re-write and re-submit, are needed for further review. 

Introduction is too short and needs more clarification.

First sentence: "excellent overall survival of 95% at  years in children with low-grade glioma" - at how many years?

  • Thank you very much for this remark, we corrected the manuscript:
  • “Long-term overall survival for children with low-grade glioma (LGG) is now of 95% at 5 years [1] and of 87% at 10 years [2] thanks to recent scientific advances.”

Are there any data on recurrence in the adult life? Please include in form of a literature review recent studies on midline gliomas. What survival - progression-free survival? Overall survival? What are the treatment strategies - resection, biopsy, radiotherapy, chemotherapy? Protocols?

  • Thank you very much for your remark, indeed our manuscript requires some precisions;

We defined the midline tumors as limited to the diencephalon, optic nerves, optic chiasm, pituitary stalk area, hypothalamus, epiphysis, thalamus, and third ventricle.

We focused on this location because it is at greater risk of endocrine sequelae, and because it is a fairly homogeneous population in terms of oncological treatments.

  • There are no scientific studies on midline LGGs as we have defined them. No series takes exactly the anatomical definition that we have chosen:

There are articles on CNS LGGs with subgroups on optic pathway gliomas for example. It is therefore difficult to specify the survival of these gliomas.

We added these elements to our manuscript, as well as your very interesting article suggestions:

While overall survival is very good, progression-free survival is less than 40% at 5 years [19]. LGG therefore often becomes a chronic disease. Over the long term, overall survival is 50.4% at 18 years for gliomas of the optic pathways in particular [19]. The main cause of death is tumor progression. The neuro-cognitive sequelae are often significant in the long term [20]. Age and intracranial hypertension at diagnosis are often associated with a worse prognosis [19]. Currently, the management of LGG is primarily surgical excision, which can be curative when total excision is possible. Unlike other LGGs in which surgery is often the only treatment (e.g., posterior fossa), surgical removal of midline LGGs is often not possible due to proximity to the optic chiasm, the trunk cerebral and hypothalamic pituitary axis [21]. Indeed, the risk of visual, neurological or endocrine sequelae is major [20]. When surgery is impossible or when residual tumor persists after surgery, the risk of progression or relapse is high. In this case, chemotherapy is usually given. The 3 most common protocols are, depending on the case, the BBSFOP protocol (6 chemotherapies are administered sequentially for 16 months and including carboplatin, procarbazine, etoposide, cisplatin, vincristine and cyclophosphamide) [22], the LGG 2004 protocol (vincristine carboplatin) [23] and the Vinblastine protocol [24]. Sometimes, the most suitable treatment remains radiotherapy: but the latter is used less and less, because it is very prone to complications, especially in very young children (secondary tumors, post-radiation angiopathy, hypothalamic-pituitary dysfunction and/or slowed cognitive development) [19].

Why do these patients have GH deficiency - is it due to tumor involvement of the sella and hypophysis? Please expand. 

  • We had detailed this on the line 92:
  • “Midline location is a statistically significant independent risk factor for growth hormone deficiency, due to proximity to the pituitary and hypothalamus [5,18]. Growth hormone (GH) deficiency is common in this location due to tumor growth, tumor removal, or radiation therapy in the hypothalamic-pituitary region”.

Materials and Methods, as well as Results - It is of utmost importance to clearly define "Midline gliomas". As a midline glioma we find in the current literature diffuse midline glioma with H3 K27M-mutant as  high-grade variant, but the authors seem to have something different in mind. What were the localizations?

  • These are low-grade gliomas located in what has been called the midline, and which we ourselves have defined by: the diencephalon, optic nerves, optic chiasm, pituitary stalk area, hypothalamus, epiphysis , thalamus, and third ventricle.
  • This is totally different from infiltrating gliomas (diffuse midline glioma H3 K27M) which are high-grade gliomas whose survival is on average 2 years. Given the very low median survival for these patients, the question of growth hormone supplementation unfortunately does not arise for these patients.

 Please cite any relevant literature, classification or definition of "low grade midline glioma".

  • As said above, there is no classification or scientific articles on midline gliomas as we have defined it ourselves.

 I suggest to include the Table with all patients and their characteristics and characteristics i.e. localization of the tumor.

  • Thank you for your comment, it would have been very interesting indeed. Unfortunately, we did not collect the exact location of the tumor during data collection and we do not have the ability to go back into patient records.

How many "non-midline" low grade pediatric gliomas were seen in the two centers in the time period?

  • We are very sorry but we do not know, and we have no way of knowing this. we focused on the hypothalamic pituitary region during our data collection.

 Oligoastrocytomas do not exist according to the new classification of WHO (2016, 2021 revision). Were there re-classifications?

  • This is a very good point, thank you very much. The histological diagnoses were all established from the 2007 WHO classification or earlier since the diagnosis had taken place before the end of 2016. They could therefore be modified today by the new WHO classification of 2016 or 2021.

There were no reclassifications. We had 4 types of tumors in our study (pilocytic astrocytoma grade I; astrocytoma grade II; oligoastrocytoma grade II; oligodendroglioma grade II).

Unfortunately, we cannot ask the anatomo-pathology laboratory to reclassify all tumor specimens according to the new WHO classification.

Where were the tumors exactly localized? What is the definition of the "midline glioma"? It would be good to have several illustrative cases with neuroradiological imaging (with/wo progression) in order to describe the tumors more nearly to the readership.

  • We defined the midline tumors as limited to the diencephalon, optic nerves, optic chiasm, pituitary stalk area, hypothalamus, epiphysis, thalamus, and third ventricle.
  • It is a very good idea to add radiological illustrations. We have added the images of the MRI before and after the initiation of patient number 106 on GHRT.

Only 57 (53.3%) of patients had surgery? - this cannot be true. How was the diagnosis secured? How many patients received biopsy?

  • 28 patients received a biopsy. 41 patients underwent complete or incomplete resection. 55 patients therefore did not have an anatomo-pathological diagnosis. Among them, 42 patients were carriers of NF1 and had a typical appearance of glioma of the optic pathways on the MRI: in this precise case, the biopsy is not necessary, because the diagnosis of GVO is highly probable and the biopsy not without risk.
  • 13 patients did not have NF1 and did not have an anatomo-pathological confirmation because the MRI was very suggestive of glioma of the optic pathways and the biopsy was too risky for their location. They were very young patients. For one of them, the biopsy had not been done because it was a therapeutic emergency to start chemotherapy.

What was the treatment strategy, which patients and why received complete resection, subtotal resection.

  • As said above, we have added these details in the introduction.

Currently, the management of LGG is primarily surgical excision, which can be curative when total excision is possible. Unlike other LGGs in which surgery is often the only treatment (e.g., posterior fossa), surgical removal of midline LGGs is often not possible due to proximity to the optic chiasm, the trunk cerebral and hypothalamic pituitary axis. Indeed, the risk of visual, neurological or endocrine sequelae is major.

When surgery is impossible or when residual tumor persists after surgery, the risk of progression or relapse is high. In this case, chemotherapy is usually given. The 3 most common protocols are, depending on the case, the BBSFOP protocol (6 chemotherapies are administered sequentially for 16 months and including carboplatin, procarbazine, etoposide, cisplatin, vincristine and cyclophosphamide), the LGG 2004 protocol (vincristine carboplatin) and the Vinblastine protocol. Sometimes, the most suitable treatment remains radiotherapy: but the latter is used less and less, because it is very prone to complications, especially in very young children (secondary tumours, post-radiation angiopathy, hypothalamic-pituitary dysfunction and/or slowed cognitive development).

What was the outcome of patients who did receive complete resection? - I suggest to include all these parameters into the analysis of survival. 

  • We have added these data to the manuscript for clarity;
  • “In our study, 9 patients had benefited from total excision. Among them, there had been no death, and only one relapse (p=0.08). Furthermore, none of the patients who could benefit from total excision had received adjuvant chemotherapy. Chemotherapy was therefore administered if the tumor was not operable, or when the resection was incomplete.”

What were the criteria for stable disease and for remission?

  • Thank you for this very good remark, we have corrected the manuscript to clarify these definitions, line 177.
  • “The tumor was considered stabilized when it had not increased in volume on two consecutive MRI scans, one year apart, without any oncological treatment. If no tumor residue was visible on the MRI one year after the end of the oncological treatments, the patient was then considered to be in complete remission.”
  • it was therefore a stable disease as defined according to local radiological assessment. Indeed, chemotherapy for example most often allows stabilization of the tumor but not its disappearance.

In how many patients can we talk about stable disease, who had a tumor remnant which did not progress?

  • Complete remission exists only in case of total surgical excision, which is rarely the case in midline tumors (only 9 patients in our study). For all the other patients, it was therefore a stable disease. Indeed, chemotherapy for example allows stabilization of the tumor but not its disappearance.

Discussion is quite disorganized, I suggest to cut it into subsection on: pediatric low grade gliomas, midline gliomas, endocrinological aspect of pediatric glioms, GH RH therapy, current studies, limitations, future advances. 

  • thank you very much for your suggestions, we have reorganized the discussion as requested

Please include and comment the current literature on the subject:

  • thank you very much for these very interesting article suggestions, we have added them to our manuscript.

Thomale UW, Gnekow AK, Kandels D, Bison B, Hernáiz Driever P, Witt O, Pietsch T, Koch A, Capper D, Kortmann RD, Timmermann B, Harrabi S, Simon M, El Damaty A, Krauss J, Schuhmann MU, Aigner A. Long-term follow-up of surgical intervention pattern in pediatric low-grade gliomas: report from the German SIOP-LGG 2004 cohort. J Neurosurg Pediatr. 2022 Jul 22:1-14. doi: 10.3171/2022.6.PEDS22108. Epub ahead of print. PMID: 35901673. Tan JY, Wijesinghe IVS, Alfarizal Kamarudin MN, Parhar I. Paediatric Gliomas: BRAF and Histone H3 as Biomarkers, Therapy and Perspective of Liquid Biopsies. Cancers (Basel). 2021 Feb 4;13(4):607. doi: 10.3390/cancers13040607. PMID: 33557011; PMCID: PMC7913734.   Traunwieser T, Kandels D, Pauls F, Pietsch T, Warmuth-Metz M, Bison B, Krauss J, Kortmann RD, Timmermann B, Thomale UW, Luettich P, Neumann-Holbeck A, Tischler T, Hernáiz Driever P, Witt O, Gnekow AK. Long-term cognitive deficits in pediatric low-grade glioma (LGG) survivors reflect pretreatment conditions-report from the German LGG studies. Neurooncol Adv. 2020 Aug 8;2(1):vdaa094. doi: 10.1093/noajnl/vdaa094. PMID: 32968720; PMCID: PMC7497816.  

Reviewer 3 Report

Authors present a bicentric retrospective study with 124 patients under the age of 18, diagnosed with a median low-grade glial tumor between 1998 and 2016. There were 17 patients who were supplemented with growth hormone (14%) and 107 patients in the unsupplemented group (86%). Relapse occurred in 65 patients (45.5%), 7 patients died (4.9%): no death  occurred in patients receiving GHRT.  There was no significant difference between the two groups since relapse concerned. Authors conclude in their observations that GHRT seems to be safe in children with midline low grade glioma, as no significant  risk of relapse has been demonstrated.

Low number of patients, retrospective character of the study as well as large discrepancy between the two groups (17 vs. 107 patients) are major drawbacks of the study, so no serious conclusion can be statistically drawn. However, this is a rare tumor and any clinical experience is welcome, however lot more clinical , neuroradiological and neuropathological data with thorough revision of the manuscript , i.e. re-write and re-submit, are needed for further review. 

Introduction is too short and needs more clarification. First sentence: "excellent overall survival of 95% at  years in children with low-grade glioma" - at how many years? Are there any data on recurrence in the adult life? Please include in form of a literature review recent studies on midline gliomas. What survival - progression-free survival? Overall survival? What are the treatment strategies - resection, biopsy, radiotherapy, chemotherapy? Protocols? Why do these patients have GH deficiency - is it due to tumor involvement of the sella and hypophysis? Please expand. 

Materials and Methods, as well as Results - It is of utmost importance to clearly define "Midline gliomas". As a midline glioma we find in the current literature diffuse midline glioma with H3 K27M-mutant as  high-grade variant, but the authors seem to have something different in mind. What were the localizations? Please cite any relevant literature, classification or definition of "low grade midline glioma". I suggest to include the Table with all patients and their characteristics and characteristics i.e. localization of the tumor. How many "non-midline" low grade pediatric gliomas were seen in the two centers in the time period?

 Oligoastrocytomas do not exist according to the new classification of WHO (2016, 2021 revision). Were there re-classifications? Where were the tumors exactly localized? What is the definition of the "midline glioma"? It would be good to have several illustrative cases with neuroradiological imaging (with/wo progression) in order to describe the tumors more nearly to the readership.

Only 57 (53.3%) of patients had surgery? - this cannot be true. How was the diagnosis secured? How many patients received biopsy? What was the treatment strategy, which patients and why received complete resection, subtotal resection. What was the outcome of patients who did receive complete resection? - I suggest to include all these parameters into the analysis of survival.  What were the criteria for stable disease and for remission? In how many patients can we talk about stable disease, who had a tumor remnant which did not progress?

Discussion is quite disorganized, I suggest to cut it into subsection on: pediatric low grade gliomas, midline gliomas, endocrinological aspect of pediatric glioms, GH RH therapy, current studies, limitations, future advances. 

Please include and comment the current literature on the subject:

Thomale UW, Gnekow AK, Kandels D, Bison B, Hernáiz Driever P, Witt O, Pietsch T, Koch A, Capper D, Kortmann RD, Timmermann B, Harrabi S, Simon M, El Damaty A, Krauss J, Schuhmann MU, Aigner A. Long-term follow-up of surgical intervention pattern in pediatric low-grade gliomas: report from the German SIOP-LGG 2004 cohort. J Neurosurg Pediatr. 2022 Jul 22:1-14. doi: 10.3171/2022.6.PEDS22108. Epub ahead of print. PMID: 35901673. Tan JY, Wijesinghe IVS, Alfarizal Kamarudin MN, Parhar I. Paediatric Gliomas: BRAF and Histone H3 as Biomarkers, Therapy and Perspective of Liquid Biopsies. Cancers (Basel). 2021 Feb 4;13(4):607. doi: 10.3390/cancers13040607. PMID: 33557011; PMCID: PMC7913734.   Traunwieser T, Kandels D, Pauls F, Pietsch T, Warmuth-Metz M, Bison B, Krauss J, Kortmann RD, Timmermann B, Thomale UW, Luettich P, Neumann-Holbeck A, Tischler T, Hernáiz Driever P, Witt O, Gnekow AK. Long-term cognitive deficits in pediatric low-grade glioma (LGG) survivors reflect pretreatment conditions-report from the German LGG studies. Neurooncol Adv. 2020 Aug 8;2(1):vdaa094. doi: 10.1093/noajnl/vdaa094. PMID: 32968720; PMCID: PMC7497816.  

Author Response

Reviewer 3 : point by point :

Growth Hormone Replacement Therapy seems to be safe in children with low-grade midline glioma: a series of 124 cases with review of the literature.

  1. Please, re-write the sentence. : Advances over the past three decades allow for excellent overall survival of 95% at years in children with low-grade glioma (LGG)

  • Thank you very much. We corrected the sentence line 53:
  • “Long-term overall survival for children with low-grade glioma (LGG) is now of 87% at 10 years thanks to recent scientific advances.”

  1. Therefore, it is important to assess the risk of tumor recurrence or progression in pa- 64 tients with LGG when GHRT is used. This question is mainly important when a tumor 65 residue is present. > Please re-structure the sentence for better understanding.

  • Thank you very much for your comment, we have reformulated the sentence like this:
  • “Therefore, it is important to assess the risk of tumor recurrence or progression in patients with LGG when GHRT is used, especially in case of residual tumor.”

  1. Is there a correlation between patient with pilocytic astrocytoma receiving chemotherapy with the patients receiving GHRT?
  • Thank you very much for your very relevant comment. Here is the detail regarding your query; the small p is equal to 1 (because no patient with an AP not treated with chemotherapy received GHRT)
  • Would you like me to add this information in the text?

  • PA = pilocytic astrocytoma

PA and Chemo

PA without chemo

with GHRT

12

0

without GHRT

51

45

Furthermore, none of the patients who could benefit from complete excision had undergone adjuvant chemotherapy. Chemotherapy was therefore administered if the tumor was not operable, or when the resection was incomplete.

  1. Does GHRT treatment correlates with the size of the tumor (quantified using MRI scan)?

  • Thank you for this suggestion, it would indeed have been very interesting. Unfortunately, we did not collect this data in the data collection.

  1. In Figure 2, the authors should do the labeling of the survival curve in English & also put the legend on the top right corner of the plot describing the blue & the red cohort.

  • Thank you very much, we have corrected this.

Round 2

Reviewer 1 Report

Thank you for your detailed edits.

Author Response

Thank you very much for your comment.

Reviewer 3 Report

Authors have answered several of the remarks. However further clarification is needed: "

  • "Thank you for your comment, it would have been very interesting indeed. Unfortunately, we did not collect the exact location of the tumor during data collection and we do not have the ability to go back into patient records." - 

This answer does not seem logical; if the manuscript is revolving around MIDLINE LGG but location of the tumor is not collected, than the manuscript losses its inner logic. Therefore I urge authors to add the requested Table with exact neuroradiological localization of the tumors. 

Only 53.3% of patients underwent surgery, please add the explanation you provided to text of the manuscript. 

How many patients did have any endocrinological disturbancy - pituitary insufficiency?

Author Response

Authors have answered several of the remarks. However further clarification is needed: "

  • "Thank you for your comment, it would have been very interesting indeed. Unfortunately, we did not collect the exact location of the tumor during data collection and we do not have the ability to go back into patient records." - 

This answer does not seem logical; if the manuscript is revolving around MIDLINE LGG but location of the tumor is not collected, than the manuscript losses its inner logic. Therefore I urge authors to add the requested Table with exact neuroradiological localization of the tumors. 

  • We went to look for the missing data concerning the localization.You will find in the new manuscript two tables (No. 2 and 3) with all the characteristics for each patient, including the location. For clarity, we have made a table with patients supplemented with GH, and a table for patients not supplemented. This table is in fact very large, we hope that this will not pose a problem for publication.
  • In addition, for greater clarity, I redid the patient numbering. Indeed, the previous numbering came from a spreadsheet which included the excluded patients (from 1 to 189). I preferred to give numbers ranging from 1 to 124. I therefore coordinated the corresponding numbers in the text.

Only 53.3% of patients underwent surgery, please add the explanation you provided to text of the manuscript. 

  • Thank you for your remark, we have added this in the manuscript as well as an associated reference.

How many patients did have any endocrinological disturbancy - pituitary insufficiency?

  • 19 patients had precocious puberty in our study. At least 16 patients had pituitary deficiency other than GH deficiency. However, there is a lot of data missing for this last piece of information.

Round 3

Reviewer 3 Report

The authors have sufficiently responded to reviewers remarks.